# Spin-dependent quantum interference in photoemission process from spin-orbit coupled states

Koichiro Yaji[1,*], Kenta Kuroda[1,*], Sogen Toyohisa[1], Ayumi Harasawa[1], Yukiaki Ishida[1], Shuntaro Watanabe[2], Chuangtian Chen[3], Katsuyoshi Kobayashi[4], Fumio Komori[1] & Shik Shin[1]

Spin–orbit interaction entangles the orbitals with the different spins. The spin–orbital-entangled states were discovered in surface states of topological insulators. However, the spin–orbital-entanglement is not specialized in the topological surface states. Here, we show the spin–orbital texture in a surface state of Bi(111) by laser-based spin- and angle-resolved photoelectron spectroscopy (laser-SARPES) and describe three-dimensional spin-rotation effect in photoemission resulting from spin-dependent quantum interference. Our model reveals that, in the spin–orbit-coupled systems, the spins pointing to the mutually opposite directions are independently locked to the orbital symmetries. Furthermore, direct detection of coherent spin phenomena by laser-SARPES enables us to clarify the phase of the dipole transition matrix element responsible for the spin direction in photoexcited states. These results permit the tuning of the spin polarization of optically excited electrons in solids with strong spin–orbit interaction.

[1] Institute for Solid State Physics, The University of Tokyo, 5-1-5 Kashiwanoha, Kashiwa, Chiba 277-8581, Japan. [2] Research Institute for Science and Technology, Tokyo University of Science, Chiba 278-8510, Japan. [3] Beijing Center for Crystal Research and Development, Chinese Academy of Science, Zhongguancun, Beijing 100190, China. [4] Department of Physics, Ochanomizu University, Tokyo 112-8610, Japan. * These authors contributed equally to this work. Correspondence and requests for materials should be addressed to K.Y. (email: yaji@issp.u-tokyo.ac.jp) or to F.K. (email: komori@issp.u-tokyo.ac.jp).

Strongly spin–orbit-coupled materials such as Rashba systems and topological insulators have been intensively studied not only because of fundamental scientific interest on unique spin textures of the surface states but also realizing spintronic devices[1–6]. In a standard model of the spin texture on the spin–orbit-coupled materials, the spin is locked to the momentum of an electron, resulting in a single-chiral spin texture[5,7]. However, this picture is incomplete to describe the spin texture of the real system. Remarkably, the entangled spin–orbital textures on a topological insulator, $Bi_2Se_3$ (refs 8–13), and a Rashba-type ternary alloy, BiTeI[14,15], were revealed experimentally and theoretically; the spin texture is locked to the orbital texture of the bands. The spin–orbital-entanglement is a general consequence of the strong spin–orbit coupling, and thus is important not only for surface states but also bulk states.

In this article, we report on the spin–orbital texture of a surface state of an elemental Bi(111), which was considered to show the single-chiral spin texture[16–18], investigated by spin- and angle-resolved photoelectron spectroscopy using a vacuum ultraviolet laser (laser-SARPES). We establish a general description of the spin–orbital texture in even–odd parity symmetry. Moreover, we draw a new concept to determine the phase of the dipole transition matrix element of photoemission through the spin-dependent quantum interference, which relies on the spin–orbital-entanglement and the laser field. We elucidate that the phase governs the spin direction in the final spinor field. The spin–orbital-entangled systems are one of the promising candidates[19] to realize the spin manipulation of optically excited electrons[20–22].

## Results

**Spin–orbital texture on a mirror plane**. All of the angle-resolved photoelectron spectroscopy (ARPES) and SARPES data were acquired with the fixed experimental geometry shown in Fig. 1a. Figure 1b displays an ARPES intensity image recorded in a $\bar{\Gamma}\bar{M}$ mirror plane on the Bi(111) surface. The spin-split surface states exhibit upward energy dispersions while the band dispersing downward from the $\bar{\Gamma}$ point is attributed to a bulk state. The results agree well with previous reports[16,17]. The laser-SARPES measurements were performed at selected $k$ cuts with the $s$- and $p$-light-polarizations as shown in Fig. 2a–h. In each light-polarization condition, the $y$ component of the spin polarization ($P_y$) is inverted with respect to the $\bar{\Gamma}$ point and the absolute values of $P_y$ at $k_1$ and $k_4$ are almost 100%. Moreover, we observed the $P_y$ reversal at each fixed $k$ point with switching the light polarization, whereas there was no spin polarization in the $x$ and $z$ directions ($P_{x,z}$) (Supplementary Fig. 1; Supplementary Note 1).

The wavefunction of the surface state can be decomposed into the symmetric ($|\psi_{even}\rangle$) and anti-symmetric ($|\psi_{odd}\rangle$) parts with respect to the mirror plane of the crystal. According to the dipole selection rule of photoemission, only the $|\psi_{odd}\rangle$ ($|\psi_{even}\rangle$) state is excited with the $s$-($p$-)polarized light. The results of the laser-SARPES indicate that each spin-polarized branch consists of the linear combination of $|\psi_{even,\uparrow}\rangle$ and $|\psi_{odd,\downarrow}\rangle$ states or $|\psi_{even,\downarrow}\rangle$ and $|\psi_{odd,\uparrow}\rangle$ states (Fig. 2i). Recent orbital-parity-based studies of the spin-polarized surface states on W(110) and $Bi_2Se_3$ also came to essentially the same conclusions[13,23,24].

To understand the reversal spin polarization in the mirror plane, we establish a model based on spinors coupled to the $|\psi_{even}\rangle$ and $|\psi_{odd}\rangle$ states. The initial states of the spin-lifted wavefunctions are denoted by

$$\Psi = \begin{pmatrix} \Psi_\uparrow \\ \Psi_\downarrow \end{pmatrix} = \begin{pmatrix} \psi_{even,\uparrow} + \psi_{odd,\uparrow} \\ \psi_{even,\downarrow} + \psi_{odd,\downarrow} \end{pmatrix}. \tag{1}$$

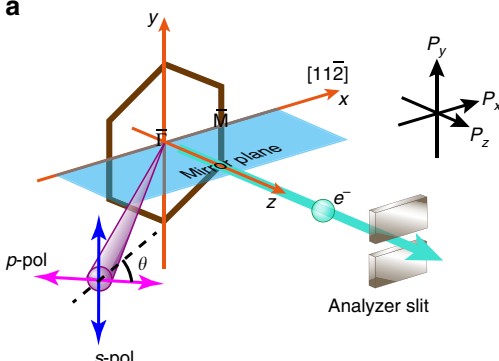

a

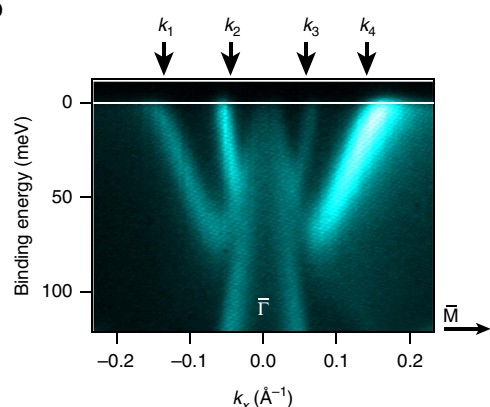

b

**Figure 1 | Experimental geometry and electronic band structure measured by ARPES.** (**a**) Schematic drawing of the experimental geometry and a surface Brillouin zone of the Bi(111) surface are represented. The experimental geometry was preserved in the present study thanks to an electron deflector function of the photoelectron analyzer. The definition of the spin polarization direction is depicted in the image. The angle between the light and the analyzer was fixed to 50°. The surface normal corresponds to the analyzer axis. The green parallelogram represents the light incident plane that is along the $\bar{\Gamma}\bar{M}$ mirror plane of the Bi(111) surface. For the $p$-($s$-) polarization, the electric-field vector of the laser is parallel (perpendicular) to the light incident plane. The electric-field vector of the linearly polarized light can be continuously rotated: the $\theta$ represents the angle between the electric-field vector of the light and the mirror plane of the Bi(111) surface. (**b**) ARPES intensity image along the $\bar{\Gamma}\bar{M}$ mirror plane on Bi(111). The ARPES data was recorded with the $p$-polarization. The spin-resolved data were measured at the wave numbers of $k_1 - k_4$.

Here, we introduce a mirror-reflection operator $\hat{M}$. Note, the spin-quantization axis is defined as a direction perpendicular to the mirror plane, which corresponds to the $y$ direction in the present system. As a consequence of the mirror operation of equation (1), we obtain the following equation;

$$\hat{M}\Psi = \begin{pmatrix} i\psi_{even,\uparrow y} & -i\psi_{odd,\uparrow y} \\ -i\psi_{even,\downarrow y} & +i\psi_{odd,\downarrow y} \end{pmatrix}. \tag{2}$$

Thus, the eigenfunctions of the mirror eigenvalues $+i$ and $-i$ are given by

$$\Psi_{+i} = \begin{pmatrix} \psi_{even,\uparrow y} \\ \psi_{odd,\downarrow y} \end{pmatrix} \text{ and } \Psi_{-i} = \begin{pmatrix} \psi_{odd,\uparrow y} \\ \psi_{even,\downarrow y} \end{pmatrix}. \tag{3}$$

From this simple calculation, we reveal that the spins pointing to the mutually opposite directions with respect to the mirror plane are locked to the even and odd parts of the spin-lifted states. This concept not only clearly explains the present results but also is generally applicable for explaining the spin–orbital texture on the

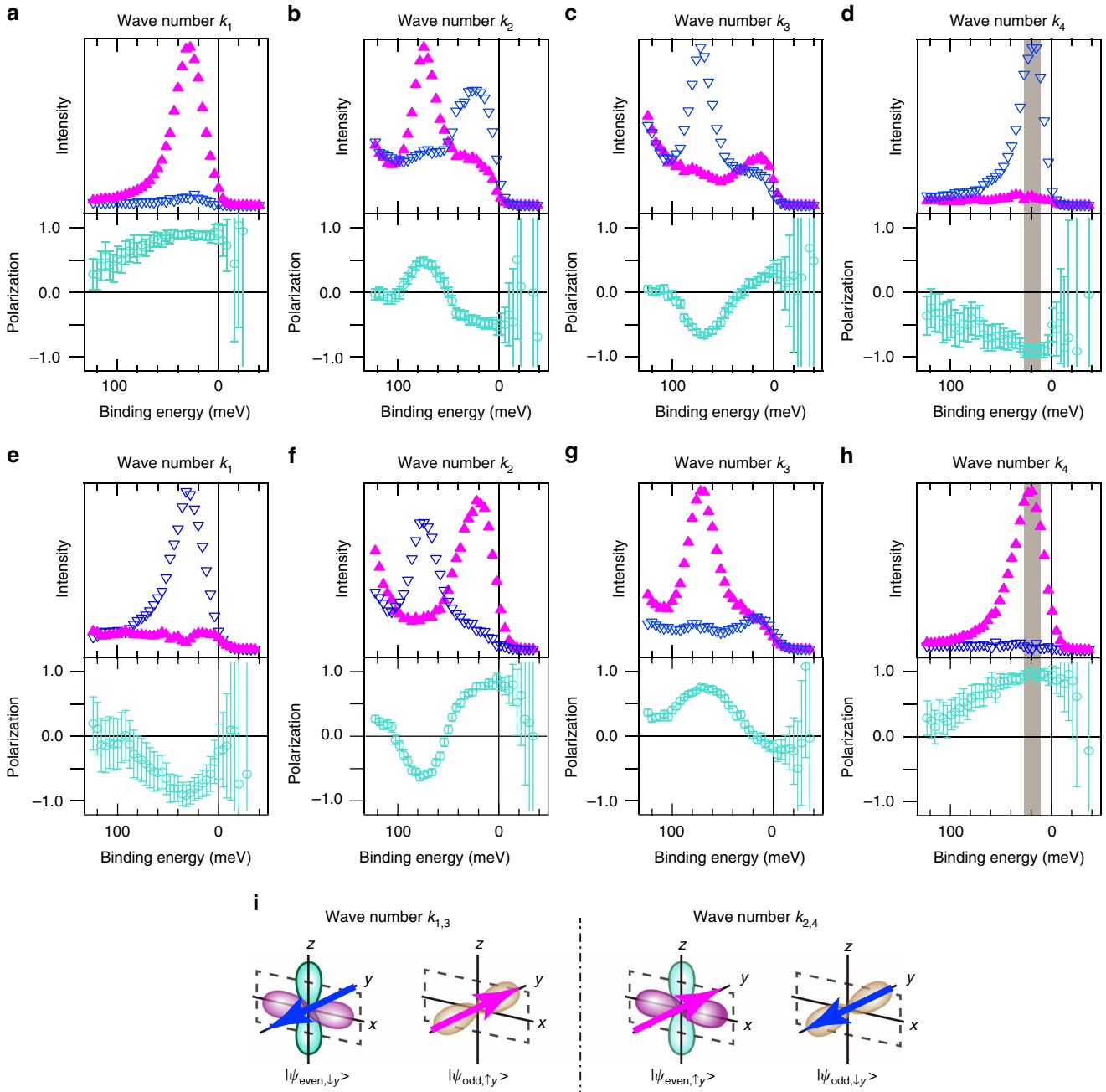

**Figure 2 | Laser-SARPES spectra of spin-polarized surface states on Bi(111).** (**a–h**) The $y$ component of spin-resolved photoelectron spectra and spin polarizations at the wave vectors of $k_1 - k_4$ shown in Fig. 1b measured with (**a–d**) $s$- and (**e–h**) $p$-polarizations are displayed. The $y$ direction is perpendicular to the $\bar{\Gamma}\bar{M}$ mirror plane. The spins pointing toward $[\bar{1}\,1\,0]$ and $[1\,\bar{1}\,0]$ directions are plotted by magenta and blue triangles, respectively. Turquoise plots represent the $y$ component of the spin polarization. The error bars represent the s.d. of the measurements. (**i**) Schematic drawing of the spin–orbital texture on the $\bar{\Gamma}\bar{M}$ mirror axis. The $p_x$ and $p_z$ states can be $|\psi_{\text{even}}\rangle$, and $p_y$ $|\psi_{\text{odd}}\rangle$.

mirror plane. The spin expectation values of these states are calculated with the Pauli matrices $\sigma_{x,y,z}$: the $y$ spin component can be finite while the $x$ and $z$ spin components are strictly 0 (Supplementary Note 2).

Now, we show the calculated band structure of the surface state on Bi(111) in Fig. 3a. Figure 3b exhibits spin expectation values of the lower surface band as a function of the wave number on $\bar{\Gamma}\bar{M}$: the spin expectation values of the $|\psi_{\text{even}}\rangle$ and $|\psi_{\text{odd}}\rangle$ states are $+1$ and $-1$, respectively. Here, the spin polarization rapidly reduced near the $\bar{\Gamma}$ point can arise from the hybridization with the bulk states. By contrast, for the upper surface band (Fig. 3c), the spin expectation values of the $|\psi_{\text{even}}\rangle$ and $|\psi_{\text{odd}}\rangle$ states are fully

reversed. These results agree with equation (3). The net spin polarizations of the surface states, represented in Fig. 3a, result from the summation of the spin expectation values with taking the weight of each spin–orbital-coupled state.

**Spin-dependent quantum interference of photoelectron.** Even if the mirror symmetry governs the spin orientation in the initial states, rotating the electric-field vector of the incident linearly polarized light can break the mirror symmetry of the experimental geometry, which leads to the spin polarization of photoelectrons in the $x$ and $z$ directions (Supplementary Fig. 2;

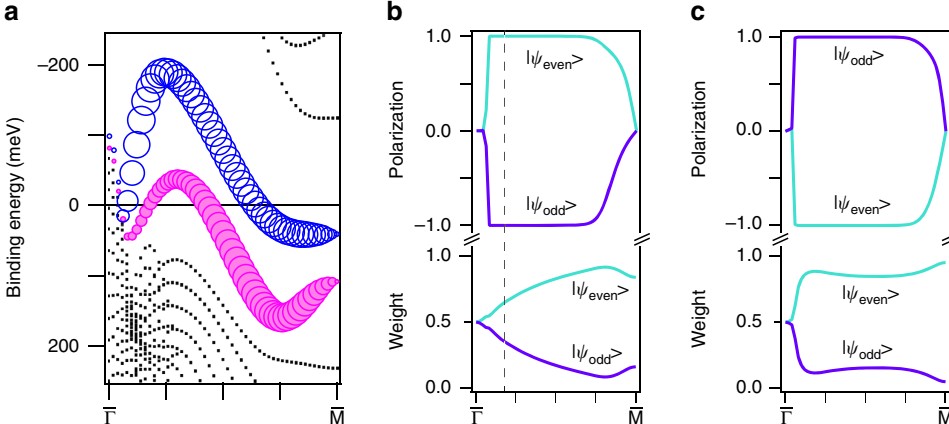

**Figure 3 | Calculated spin polarizations and weights of the wavefunctions.** (**a**) Calculated band structure for a free-standing 30-bilayer Bi slab along $\bar{\Gamma}\bar{M}$. The magenta (blue) circles represent the surface state with the spin direction pointing to the $[\bar{1}10]$ ($[1\bar{1}0]$) direction. The size of the circles is proportional to the absolute value of the net spin polarization. (**b,c**) Calculated spin expectation values and the weights of the even and odd parity components of the wavefunctions for the lower (**b**) and upper (**c**) surface states as a function of the wave number along $\bar{\Gamma}\bar{M}$. The dashed line in (**b**) corresponds to the $k_4$ point.

Supplementary Note 1). This is produced by the spin-dependent interference of the wavefunctions, resulting from simultaneous excitation of the $|\psi_{\text{even},\uparrow y}\rangle$ and $|\psi_{\text{odd},\downarrow y}\rangle$ ($|\psi_{\text{odd},\uparrow y}\rangle$ and $|\psi_{\text{even},\downarrow y}\rangle$) states. When we consider the eigenstates $|\Psi_{+i}\rangle$, the spin polarization $P_{x,y,z}$ of photoelectrons and the photoelectron intensity $I_{\text{total}}$ are expressed as a function of the light-polarization angle ($\theta$) from the mirror plane (Fig. 1a) as (Supplementary Note 2)

$$P_x = \frac{2u\sin\alpha\tan\theta}{1+u^2\tan^2\theta},$$
$$P_y = \frac{1-u^2\tan^2\theta}{1+u^2\tan^2\theta},$$
$$P_z = \frac{2u\cos\alpha\tan\theta}{1+u^2\tan^2\theta},$$
$$\frac{I_{\text{total}}(\theta)}{I_{\text{total}}(\theta=0)} = \cos^2\theta + u^2\sin^2\theta,$$

with the following ratio between the dipole matrix elements in photoemission:

$$\frac{\langle\psi_{\text{final}}|\mathbf{A}\cdot\mathbf{p}|\psi_{\text{odd},\downarrow y}\rangle}{\langle\psi_{\text{final}}|\mathbf{A}\cdot\mathbf{p}|\psi_{\text{even},\uparrow y}\rangle} = u\mathrm{e}^{i\alpha}. \qquad (5)$$

Here, $\mathbf{A}$ is the vector potential of the light, $\mathbf{p}$ the momentum operator and $\psi_{\text{final}}$ a final-state wavefunction that is assumed to be spin-degenerated for simplicity. Then, $\alpha$ represents a phase difference between the dipole matrix elements from the even and odd states, and $u$ is an absolute value of the complex number. As a consequence of the equation (4), the signs of $P_x$ and $P_z$ can be classified into four classes depending on the value of $\alpha$ (Fig. 4a–d).

To demonstrate the above prospect, we show the observed $\theta$ dependence of $P_{x,y,z}$ at $k_4$ in Fig. 4e,f since the $|\psi_{\text{even},\uparrow y}\rangle$ and $|\psi_{\text{odd},\downarrow y}\rangle$ states exhibit the 100% spin polarization in the initial states. The $P_{x,y,z}$ oscillate as a function of $\theta$ as expected. The $P_y$ is to be zero at $\theta \sim 60°$ and $120°$, indicating that photoelectrons from the $|\psi_{\text{even},\uparrow y}\rangle$ and $|\psi_{\text{odd},\downarrow y}\rangle$ states cancel out each other at these angles. By contrast, the $|P_{x,z}|$ are almost zero at $\theta \sim 0°$, $90°$ and $180°$, and exhibit maximum values at $\theta \sim 60°$ and $120°$. The signs of $P_x$ and $P_z$ are negative (positive) with $0 < \theta < 90°$ ($90° < \theta < 180°$). Thus, we can immediately judge $\pi < \alpha < 3\pi/2$ at $k_4$ using Fig. 4a–d. The experimental results of $P_{x,y,z}$ and the intensity were well reproduced by the equation (4) with $u = 0.62$

and $\alpha = 1.3\pi$. Here, we note that the $\theta$ dependence of $P_{x,y,z}$ should be changed with changing the photon energy since the spin-dependent matrix elements, that is, the spin-dependent matrix elements, are different.

## Discussion

The electron–photon interaction Hamiltonian of photoemission is given by the three terms corresponding to the dipole transition, surface photoemission, and spin–orbit coupling[25]. In the earlier theoretical work[26], the spin rotation effect in photoemission was discussed with both spin-conserving and spin-flipping transitions with employing the dipole transition and spin–orbit terms in the interaction Hamiltonian. Subsequently, Jozwiak et al.[27] experimentally demonstrated that the spin polarization of photoelectrons from the surface state of $Bi_2Se_3$ is largely changed compared with that of the initial state, which was explained by the spin-flip transition in photoemission: they considered the average spin texture in the initial state, but not the spin–orbital texture. In the present study, we demonstrate that the spin polarization of the photoelectrons excited by the linearly polarized light is successfully explained only with the dipole transition term in the interaction Hamiltonian with taking the mirror symmetry and the spin–orbital texture into account. It has not yet been established how important the spin rotation contribution to photoexcitation arising from the spin–orbit term is. In fact, Wissing et al.[28] pointed out that the relativistic corrections of the dipole operator would be negligibly small corrections to the spin polarization of the photoelectrons, while in the photoemission study using the circularly polarized light it was discussed that the spin–orbit term in the interaction Hamiltonian is generally strong for systems with heavy elements[25].

For another mechanism of the spin rotation, a layer-dependent interference effect in photoemission process was proposed[13]. This mechanism is only achievable in the system with the layer-dependent spin–orbital texture, and realizes the spin control of photoelectrons only by varying photon energy. The present concept is essentially different from this scheme. The spin rotation over three dimension results from simultaneous optical excitation of the linearly-combined even and odd parts of the wavefunctions, and thus the spin direction of photoelectron can be readily controlled just by tuning linear-polarization axis of the light with the fixed photon energy. This concept is comprehensive

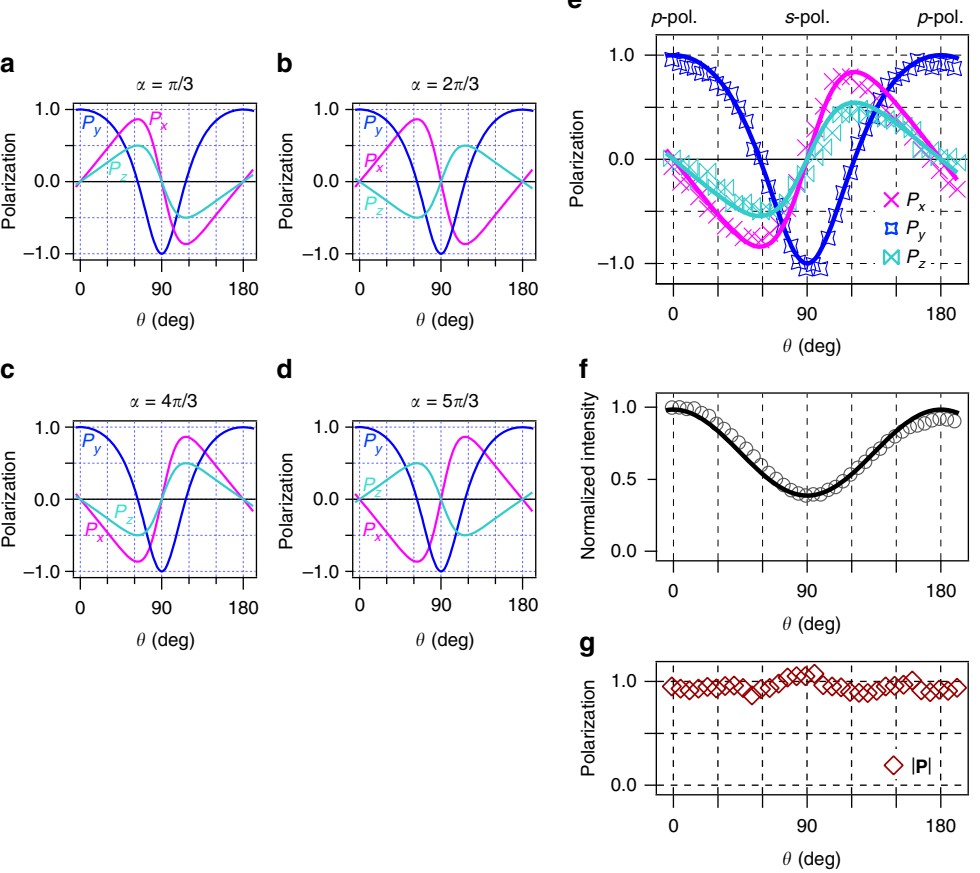

**Figure 4 | Simulated and experimental spin polarizations of photoelectrons.** (**a–d**) The phase (α in equation (5)) dependence of the spin polarization of photoelectrons based on the equation (4). The curves represent the $x$, $y$ and $z$ components of the spin polarization. In this simulation, we tentatively used $u = 0.5$ and $α = nπ/3$ ($n = 1, 2, 4, 5$). (**e–g**) The three-dimensional spin polarization (**e**), total intensity (**f**) and the magnitude of the spin polarization ($|\mathbf{P}| = \sqrt{P_x^2 + P_y^2 + P_z^2}$) (**g**) of photoelectrons at $k_4$ are plotted as a function of $θ$. In these plots, the $P_{x,y,z}$ are averaged and the photoelectron intensities are integrated in binding energy corresponding to the shaded region in Fig. 2d,h. The $θ$ is defined as an angle between the electric-field vector of the incident light and the light incident plane, meaning that the $p$-polarization ($s$-polarization) corresponds to $θ = 0°$ and 180° ($θ = 90°$).

and no longer needs the layer-dependent interference picture to demonstrate the optical spin control.

Furthermore, the present concept is applicable not only to the present system but to the other $|\psi_{\text{even}}\rangle$ and $|\psi_{\text{odd}}\rangle$ mixed systems. In the case of Bi$_2$Se$_3$, the sign of $P_x$ is the same as Bi(111), but the sign of $P_z$ opposite[19]. Thus, the phase difference of the matrix elements should be in the range of $3π/2 < α < 2π$ in accordance with Fig. 4d: the $P_{x,y,z}$ were well fitted with $u = 0.45$ and $α = 1.6π$. This indicates that the phase difference is a material-inherent variable.

So far, in the photoelectron spectroscopy, one has observed only the intensity of photoelectrons, meaning that the phase information of the dipole matrix element has been lost. By contrast, the three-dimensional SARPES with varying linear-polarization angle provides the phase information that is essential to describe the nature of the spin polarization of the photoexcited electron. The combination of three-dimensional SARPES and the linear-polarization-controlled laser is an innovative tool for quantum-mechanical understanding of the photoexcitation process.

The results offer opportunities for photocathodes as highly spin-polarized electron sources. The disadvantage of commonly used GaAs photocathodes as spin-polarized electron sources is that it is hard to tune the direction of the spin polarization and that the degree of spin polarization is only 50% (ref. 29). On the

other hand, the present expermental results clearly show the 100% spin polarization of photoelectron (Fig. 4g), as theoretically predicted in the former report[26], and its direction readily controllable just by tuning the linear photon polarization. A technique using the quantum-mechanical phase degree of freedom opens new avenues for the optical spin control.

## Methods

**Sample preparation.** The Bi sample was *in situ* prepared in a molecular beam epitaxy chamber connected to the analysis chamber. We used $n$-type Si(111) substrates. A clean Si(111) surface was prepared by flushing at 1,420 K. Then, Bi with the thickness of 100 bilayers (BL) was deposited onto the clean Si(111)-7 × 7 surface at room temperature from a Knudsen cell[30]. The deposition rate was calibrated by observing well-known quantum-well-states on the Bi film by ARPES[31]. The Bi film exhibits a sharp (1 × 1) low-energy electron-diffraction pattern and an excellent Fermi surface image by ARPES.

**Laser-ARPES and SARPES measurements.** Our ARPES and SARPES measurements using an ultraviolet laser were performed at the Institute for Solid State Physics, The University of Tokyo[32]. Our laser system provides 6.994-eV photons[33]. Photoelectrons were analysed with a combination of a ScientaOmicron DA30L analyzer and twin very-low-energy-electron-diffraction (VLEED) type spin detectors. The experimental geometry is represented in Fig. 1a. The light incident plane is in the $x$–$z$ plane on the sample axis, which corresponds to the $\overline{\Gamma}\overline{M}$ mirror plane. We used linearly polarized light, and the direction of its electric-field vector is arbitrarily adjustable between the $p$- and $s$-polarizations. Rotation angle of the electric-field vector is given by $θ$, where the light is of the $p$-($s$-)polarization at $θ = 0°$ and 180 (90°). The energy and angular resolutions were set to 6 meV

and 0.7°, respectively. The sample temperature was kept at 15 K during the laser-SARPES measurements.

**Electronic band structure calculation.** The first-principles calculation was performed using the Vienna *Ab initio* Simulation Package (VASP)[34]. The projector augmented wave (PAW) method[35] was used in the plane-wave calculation. The generalized gradient approximation (GGA) by Perdew, Burke and Ernzerhof (PBE)[36] was used for the exchange-correlation potential. The spin–orbit interaction was included. The cut-off energy was 110 eV. The Bi film was modelled by a free-standing 30-BL Bi(111) slab. The slabs in the repeated slab structure were separated by vaccums with a thickness more than 10 Å. Atom positions in the slab were taken from the experimental data shown in ref. 37.

**Data availability.** The data supporting the findings of this study are available from the corresponding author on request.

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

## Acknowledgements

Numerical calculations were performed using supercomputers at the Institute of Solid State Physics, The University of Tokyo. This work was supported by J.S.P.S. Grant-in-Aid for Scientific Research (B), Grant No. 26287061, for Scientific Research (C), Grant No. 26390063 and for Young Scientists (B) Grant No. 15K17675.

## Author contributions

K.Y. and K.Ku. conceived the research with guidance from F.K. K.Y. and S.T. fabricated and characterized the sample. K.Y. and K.Ku. carried out ARPES and SARPES measurements under the support of S.T., A.H., Y.I., S.W., C.C., F.K. and S.S. K.Ko. carried out the theoretical calculation. K.Y., K.Ku., K.Ko. and F.K. wrote the manuscript. F.K. and S.S. supervised the project. All authors discussed the paper.

## Additional information

**Competing financial interests:** The authors declare no competing financial interests.

**Publisher's note**: 

