## [Peer Review File · Nature Communications]

Reviewers' comments:

Reviewer #1 (Remarks to the Author):

Yagi et al. show a nice spin- and angle-resolved photoelectron spectroscopy study of the Bi(111) surface state which hosts Rashba spin splitting. The data in Fig. 4 is very impressive. Their main conclusion is that the wavefunction can be described by linear combination of symmetric and antisymmetric parts with opposite spin. This is interesting finding but is also found previously in other systems. On p.3, authors even write "Recent orbital-parity-based studies of the spin-polarized surface states on W(110) and Bi₂Se₃ also came to essentially the same conclusions [6, 24, 25]." The authors do make further developments on the theory of the photoemission process and how this shows up the spin interference but I unfortunately do not see this as the major conceptual advance necessary to publish this work in Nature Communications. It seems more appropriate to specialist journal e.g. Phys. Rev. B.

Further technical comments:

In Fig. 1, the measured dispersion at positive and negative k looks very different. What is the cause for this?

In Fig. 2, there seem to be large finite polarized background in some EDCs. e.g. (a) s-pol. Do the authors understand the origin of this?

Can the authors comment more on final state effects? Would their observation be photon energy dependent? Even although they do not need a multiple layer model in their picture to get the interference the surface state wave function will have some finite value on the sub surface layer.

Reviewer #2 (Remarks to the Author):

The authors have reported spin-ARPES results on Bi (111) surface. They have beautifully demonstrated that the direction in 2D sphere of the spin polarization of photoelectrons can be largely tuned by modulating the (linear) photon polarization. They also explained their experimental results for photoemission of electrons whose Bloch wavevector is on Gamma-M by using a simple model (based on the mirror symmetry) with two fitting parameters. Although similar attempts have been made for other materials including topological insulators, the experimental data presented in this paper are of higher quality and this paper has achieved one big step toward the spin modulation of photoelectrons by tuning the light polarization. Considering that the results have an important implication in developing spin-polarized electron sources and also that their experimental data is of very high quality, I find that the paper deserves publication in Nature Communications. In the following, I have some suggestions for the authors to address before publication of the paper.

The paper is currently misleading regarding the spin-flipping and photoexcitation described in Refs. [26] (Park and Louie) and [27] (Jozwiak et al.); What is meant in Ref. [27] by "spin-flipping" is that the spin polarization of photoelectrons being antiparallel to the "average" spin polarization of the initial Bloch state. Although Ref. [26] discusses both spin-conserving and spin-flipping transitions upon photoexcitation (i.e., real transition from spin up state to either spin up or spin down final electronic state), (i) Ref. [26] did not claim that spin-flipping transition is dominant and (ii) the meaning of 'spin-flipping' in Ref. [27] is different from that in Ref. [26]. In addition, considering another study by C.-Z. Xu et al., Phys. Rev. Lett. 115, 016801 (2015), it is still not established how important the spin-flipping contribution to photoexcitation arising from spin-orbit coupling is.

The disadvantage of commonly used GaAs photocathodes as spin-polarized electron sources is that it is hard to tune the direction of the spin polarization and that the degree of spin polarization is only 50% (unless in-plane strain, which makes it even harder to tune the direction of the spin polarization of photoelectrons, is used). Therefore, it would be important if the authors show that the degree of spin polarization can be much higher than that of GaAs photocathodes, i.e., 50%. Could the authors also show the magnitude of spin polarization as well when they show each (x, y, and z) component of the spin polarization in Fig. 4? It was predicted that this quantity could become almost fully polarized (100%) in Ref. [26]. If the results of the experiment actually indicates this 100% spin polarization, the results would also be important from spintronics point of view.

In Eq. (5), they have used the \hat{r} operator for optical transition matrix elements. However, \hat{r} operator is not well defined in (semi) infinite system. They should replace \hat{r} with the correct operator, i.e., the velocity operator.

Reviewer #3 (Remarks to the Author):

Yaji et al. present spin-resolved ARPES data from the Bi(111) surface. Using linearly polarized light with varying angle of the polarization with respect to the experimental mirror plane, they find that the measured spin polarization at a fixed k-point is a strong function of the light polarization and can be reversed under suitable conditions. These findings are largely identical to earlier reports on topological surface states (Refs. 2-6, 27). Contrary to what the authors suggest, there is therefore little new experimental information here, especially when one considers the well known analogy between Rashba split surface states (as on Bi(111)) and those of topological insulators (see e.g. Yan et al., Nat. Comm. 6, 10167).

The main advance of the present work, as far as I can see, is the development of a simple and transparent model to describe the photoelectron spin polarization in spin-orbit coupled systems. This model is in excellent agreement with the data and appears to be broadly applicable to different material systems including topological insulators. This advance is significant for several reasons. First, a number of earlier studies of the same effect on topological insulators did not result in a consistent picture. The present work seems to resolve this issue. Second, and arguably even more important, the present work demonstrates clearly that in spin-orbit coupled systems, measured vectorial spin polarizations do not represent the initial state but are determined by spin dependent matrix elements. While this basic insight is perhaps not unexpected for experts it is far too often ignored by the community. Finally, the work shows that strongly spin dependent matrix elements are a generic property of spin-orbit coupled systems rather than a peculiarity of topological insulators. Again, this is not surprising but largely ignored by several groups working on spin-resolved ARPES. I thus tend to support publication of this paper in Nature Communications. Spin-resolved ARPES rapidly gained prominence since the discovery of topological insulators but has been plagued by experimental difficulties and wrong interpretations of the experimental results. I expect that the present paper will contribute strongly towards a more appropriate interpretation of spin-resolved ARPES data on many topological systems, which would constitute a significant advance.

Minor comments:

1) The authors suggest repeatedly that spin-orbital entanglement would only be relevant in spin-split surface states, while, in fact, the concept is far broader and important for any system where spin-orbit interaction becomes comparable to the crystal field splitting.

1) Fig. 2 might be clearer without the spin-polarization curves.

3) The last sentence of the abstract is not very clear. Maybe the authors could say explicitly that these results permit the tuning of the spin polarization of optically excited electrons in solids with strong spin-orbit interaction.

REVIEWERS' COMMENTS:

Reviewer #1 (Remarks to the Author):

Yogi and colleague have responded well to the technical concern and I think this is a nice paper. The main advance is the new model as remarked by referee 3 which is a good simple picture. Even though, I believe that this does not move significantly beyond the picture in [13] which need not have layer-dependent effect by necessity but only multiple orbital which underpin the $\psi_{\text{even/odd}}$ picture. As a result of this I think the truly qualitative advance is still not seen here and so cannot recommend to publish in Nature Communications.

Reviewer #2 (Remarks to the Author):

I find that the authors have thoroughly addressed all the issues raised by all three referees. All three referees agree on the high quality of the research report in the paper. There is no correct / wrong answer when it comes to how important a paper is. For the same reason as in my first report, and also for the additional reason as in the report of the third referee, I believe the paper will draw attention of the general readership of Nature Communication; the paper will foster further research activities in the field of spin ARPES on several interesting / newly-discovered materials with strong spin-orbit coupling. For this reason, I recommend publication of this paper in Nature Communications in current form.

Reviewer #3 (Remarks to the Author):

I note that all three referees had only minor technical comments. These comments have been fully addressed by the authors in the revised version and the rebuttal letter.

The most substantial criticism was raised by referee 1, pointing out that the observation of spin-orbital texture as well as the decomposition of a spin-orbital entangled wave function into symmetric and antisymmetric parts with opposite spin are not new at all. While this is of course true, I believe that it misses the key-advance of this paper. What is new here is neither the concept of spin-orbital entanglement nor the experimental evidence for it but the development of a simple and transparent model to describe the spin polarization in photoemission from spin-orbit coupled systems. As described in my first report, I consider this a significant and timely advance.

Reviewer #1 (Remarks to the Author):

Yagi et al. show a nice spin- and angle-resolved photoelectron spectroscopy study of the Bi(111) surface state which hosts Rashba spin splitting. The data in Fig. 4 is very impressive. Their main conclusion is that the wavefunction can be described by linear combination of symmetric and antisymmetric parts with opposite spin. This is interesting finding but is also found previously in other systems. On p.3, authors even write “Recent orbital-parity-based studies of the spin-polarized surface states on W(110) and Bi₂Se₃ also came to essentially the same conclusions [6, 24, 25].” The authors do make further developments on the theory of the photoemission process and how this shows up the spin interference but I unfortunately do not see this as the major conceptual advance necessary to publish this work in Nature Communications. It seem more appropriate to specialist journal e.g. Phys. Rev. B.

Re) We appreciate the reviewer's careful reading and giving useful comments, which are significant to improve the paper. The reviewer #1 made a criticism on the conceptual advance. The reviewer #1 said that the spin-orbital texture has been also found previously in other systems. This is true. However, the advantage of the present study is that the physical origin of the spin-orbital texture is explicitly described based on the mirror symmetry, which is broadly applicable to other strongly spin-orbit coupled systems. More importantly, the present study holds a significant advance, that is, a proposal of a universal model to describe the photoelectron spin polarization in spin-orbit coupled systems. Nowadays, a SARPES technique is extensively utilized to investigate the spin-polarized bands in strongly spin-orbit coupled materials such as Rashba systems, topological insulators, Weyl semimetals and so on. However, there are experimental difficulties and wrong interpretations of the experimental SARPES results. The present results would largely contribute to correct understanding of the experimental SARPES results. Besides, the present concept is not limited in the fields of photoemission spectroscopy and electronic physics but would be important for spintronic technology, spin-polarized photocathodes for example. We are sure that the present work will attract great interest of the readers of the *Nature Communications*, and thus the paper is worth publishing in this journal.

Next, we make point-by-point responses.

Further technical comments:

In Fig. 1, the measured dispersion at positive and negative k looks very different. What is the cause for this?

Re) As pointed out by the reviewer #1, the photoelectron intensity at the positive and negative k is different. This intensity modulation results from the geometry effect of the photoemission experiments. In the initial state, the band dispersion of the surface state at the positive and negative k should be symmetric. However, in our experiments, the incident angle of the light is 50° with respect to the surface normal, meaning that the experimental geometry breaks the symmetry. This situation can produce the intensity difference at the positive and negative k . We expect that the intensity modulation switches when the incident angle of the light is -50° and the intensity would be symmetric when we consider the light incidence normal to the surface.

In Fig. 2, there seem to be large finite polarized background in some EDCs. e.g. (a) s-pol. Do the authors understand the origin of this?

Re) The reviewer #1 asked the origin of the spin-polarized background. To answer this question, we show typical spin-resolved EDCs and spin polarizations in a wide range in the right figure, corresponding to the results shown in Fig. 2(a,d). We find that the spin polarization is 0 at the binding energy of 150 meV while there are finite spin polarization in other energy regions.

The surface-state peaks have asymmetric tails on the higher binding energy side, caused by inelastic scattering of the photoelectrons from the spin-polarized surface states. The EDCs also exhibit large spin-polarized peaks around 220 meV, which are attributed to the spin-polarized bulk continuum [Kimura et al. PRL 105, 076804 (2010)]. These are the origin of the spin-polarized backgrounds of the EDCs found in Fig. 2.

Can the authors comment more on final state effects? Would their observation be photon energy dependent? Even although they do not need a multiple layer model in their picture to get the interference the surface state wave function will have some finite value on the sub surface layer.

Re) In the first half of the comment, the reviewer #1 asked the final state effects and the photon energy dependence. Our model of the optical spin response is based on the spin-dependent matrix elements of photoemission. Thus, the final state effects should be

significant, meaning that the θ dependence of $P_{x,y,z}$ of photoelectrons would be changed with changing the photon energy. To address this issue clearly, we added a following sentence in the end of the results section.

“Here, we note that the θ dependence of $P_{x,y,z}$ should be changed with changing the photon energy since the photoexcited states, i.e. the spin-dependent matrix elements, are different.”

In the latter half, the reviewer #1 made a comment on the multiple layer model of the spin-orbital texture. As pointed out by the reviewer #1, the surface state gradually decays into the bulk. Thus, the wavefunctions of the surface state can be expressed as a linear combination of the layer-by-layer symmetric or anti-symmetric wavefunctions. Our model simply treats the “linearly-combined” symmetric and anti-symmetric wavefunctions, meaning that the model includes the layer-by-layer picture comprehensively. Consequently, we state that our model is essential and universal, and no longer needs to consider the layer-by-layer spin-orbital texture as in the literature.

Reviewer #2 (Remarks to the Author):

The authors have reported spin-ARPES results on Bi (111) surface. They have beautifully demonstrated that the direction in 2D sphere of the spin polarization of photoelectrons can be largely tuned by modulating the (linear) photon polarization. They also explained their experimental results for photoemission of electrons whose Bloch wavevector is on Gamma-M by using a simple model (based on the mirror symmetry) with two fitting parameters. Although similar attempts have been made for other materials including topological insulators, the experimental data presented in this paper are of higher quality and this paper has achieved one big step toward the spin modulation of photoelectrons by tuning the light polarization. Considering that the results have an important implication in developing spin-polarized electron sources and also that their experimental data is of very high quality, I find that the paper deserves publication in Nature Communications. In the following, I have some suggestions for the authors to address before publication of the paper.

Re) We appreciate the reviewer’s careful reading and giving useful comments. We are also pleased that the reviewer #2 appreciates our work as “the results have an important implication in developing spin-polarized electron sources and also that their experimental data is of very high quality”, and finally made a positive suggestion for publication. Next, we make point-by-point responses.

The paper is currently misleading regarding the spin-flipping and photoexcitation described in Refs. [26] (Park and Louie) and [27] (Jozwiak et al.); What is meant in Ref. [27] by “spin-flipping” is that the spin polarization of photoelectrons being antiparallel to the “average” spin polarization of the initial Bloch state. Although Ref. [26] discusses both spin-conserving and spin-flipping transitions upon photoexcitation (i.e., real transition from spin up state to either spin up or spin down final electronic state), (i) Ref. [26] did not claim that spin-flipping transition is dominant and (ii) the meaning of ‘spin-flipping’ in Ref. [27] is different from that in Ref. [26]. In addition, considering another study by C.-Z. Xu et al., Phys. Rev. Lett. 115, 016801 (2015), it is still not established how important the spin-flipping contribution to photoexcitation arising from spin-orbit coupling is.

Re) We sincerely thank the reviewer’s expert comment that is quite helpful for us to understand the former works correctly. In accordance with this comment, we have revised one of the paragraphs in the discussion section as follows:

“The electron-photon interaction Hamiltonian of photoemission is given by the three terms corresponding to the dipole transition, surface photoemission, and spin-orbit coupling [25]. In the earlier theoretical work [26], the spin rotation effect in photoemission was discussed with both spin-conserving and spin-flipping transitions with employing the dipole transition and spin-orbit terms in the interaction Hamiltonian. Subsequently, Jozwiak *et al.* experimentally demonstrated that the spin polarization of photoelectrons from the surface state of Bi₂Se₃ is largely changed compared with that of the initial state, which was explained by the spin-flip transition in photoemission [27]: They considered the average spin texture in the initial state, but not the spin-orbital texture. In the present study, we demonstrate that the spin polarization of the photoelectrons excited by the linearly polarized light is successfully explained only with the dipole transition term in the interaction Hamiltonian with taking the mirror symmetry and the spin-orbital texture into account. It has not yet been established how important the spin rotation contribution to photoexcitation arising from the spin-orbit term is. In fact, Wissing et al. pointed out that the relativistic corrections of the dipole operator would be negligibly small corrections to the spin polarization of the photoelectrons [28], while in the photoemission study using the circularly polarized light it was discussed that the spin-orbit term in the interaction Hamiltonian is generally strong for systems with heavy elements [25].”

The disadvantage of commonly used GaAs photocathodes as spin-polarized electron sources is that it is hard to tune the direction of the spin polarization and that the degree of spin polarization is only 50% (unless in-plane strain, which makes it even harder to tune the direction

of the spin polarization of photoelectrons, is used). Therefore, it would be important if the authors show that the degree of spin polarization can be much higher than that of GaAs photocathodes, i.e., 50%. Could the authors also show the magnitude of spin polarization as well when they show each (x, y, and z) component of the spin polarization in Fig. 4? It was predicted that this quantity could become almost fully polarized (100%) in Ref. [26]. If the results of the experiment actually indicates this 100% spin polarization, the results would also be important from spintronics point of view.

Re) Thank you very much for the useful comment. We agree that it is important to show the total spin polarization of photoelectrons and to discuss possibility for the spin-polarized photocathodes. To address this issue, we added the total spin polarization in Fig. 4 and we also added a new paragraph in the end of the discussion section.

“The results offer opportunities for photocathodes as highly spin-polarized electron sources. The disadvantage of commonly used GaAs photocathodes as spin-polarized electron sources is that it is hard to tune the direction of the spin polarization and that the degree of spin polarization is only 50% [29]. On the other hand, the present experimental results clearly show the 100% spin polarization of photoelectron [see the middle panel of Fig. 4(b)], as theoretically predicted in the former report [26], and its direction readily controllable just by tuning the linear photon polarization. A technique using the quantum-mechanical phase degree of freedom opens new avenues for the optical spin control.”

In Eq. (5), they have used the ‘r’ operator for optical transition matrix elements. However, ‘r’ operator is not well defined in (semi) infinite system. They should replace ‘r’ with the correct operator, i.e., the velocity operator.

Re) In the revision, we use the $\mathbf{A} \cdot \mathbf{p}$ operator for optical transition matrix elements instead of $\mathbf{E} \cdot \mathbf{r}$.

Reviewer #3 (Remarks to the Author):

Yaji et al. present spin-resolved ARPES data from the Bi(111) surface. Using linearly polarized light with varying angle of the polarization with respect to the experimental mirror plane, they find that the measured spin polarization at a fixed k-point is a strong function of the light polarization and can be reversed under suitable conditions. These findings are largely identical to earlier reports on topological surface states (Refs. 2-6, 27). Contrary to what the authors

suggest, there is therefore little new experimental information here, especially when one considers the well known analogy between Rashba split surface states (as on Bi(111)) and those of topological insulators (see e.g. Yan et al., Nat. Comm. 6, 10167).

The main advance of the present work, as far as I can see, is the development of a simple and transparent model to describe the photoelectron spin polarization in spin-orbit coupled systems. This model is in excellent agreement with the data and appears to be broadly applicable to different material systems including topological insulators. This advance is significant for several reasons. First, a number of earlier studies of the same effect on topological insulators did not result in a consistent picture. The present work seems to resolve this issue. Second, and arguably even more important, the present work demonstrates clearly that in spin-orbit coupled systems, measured vectorial spin polarizations do not represent the initial state but are determined by spin dependent matrix elements. While this basic insight is perhaps not unexpected for experts it is far too often ignored by the community. Finally, the work shows that strongly spin dependent matrix elements are a generic property of spin-orbit coupled systems rather than a peculiarity of topological insulators. Again, this is not surprising but largely ignored by several groups working on spin-resolved ARPES. I thus tend to support publication of this paper in Nature Communications. Spin-resolved ARPES rapidly gained prominence since the discovery of topological insulators but has been plagued by experimental difficulties and wrong interpretations of the experimental results. I expect that the present paper will contribute strongly towards a more appropriate interpretation of spin-resolved ARPES data on many topical systems, which would constitute a significant advance.

Re) We appreciate the referee's careful reading and evaluating our research. We are also grateful to the reviewer's expert comments to understand the importance of our work. Finally, he/she made a favorable comment by stating "I expect that the present paper will contribute strongly towards a more appropriate interpretation of spin-resolved ARPES data on many topical systems, which would constitute a significant advance." The reviewer #3 made three minor comments. We make point-by-point responses as follows.

Minor comments:

1) *The authors suggest repeatedly that spin-orbital entanglement would only be relevant in spin-split surface states, while, in fact, the concept is far broader and important for any system where spin-orbit interaction becomes comparable to the crystal field splitting.*

Re) Thank you very much for the useful comment. As pointed out by the reviewer #3, the spin-orbital entanglement can broadly appear in strong spin-orbit coupled systems. To address this point, we added a following sentence in the introduction section.

“The spin-orbital entanglement is a general consequence of the strong spin-orbit coupling, and thus is important not only for surface states but also bulk states.”

2) *Fig. 2 might be clearer without the spin-polarization curves.*

Re) We agree with this comment. In the revision, we display the spin-resolved spectra and the spin polarizations separately.

3) *The last sentence of the abstract is not very clear. Maybe the authors could say explicitly that there results permit the tuning of the spin polarization of optically excited electrons in solids with strong spin-orbit interaction.*

Re) Thank you very much for the useful suggestion. We have revised the sentence in the abstract.

Reviewer #1 (Remarks to the Author):

Yogi and colleague have responded well to the technical concern and I think this is a nice paper. The main advance is the new model as remarked by referee 3 which is a good simple picture. Even though, I believe that this does not move significantly beyond the picture in [13] which need not have layer-dependent effect by necessity but only multiple orbital which underpin the $\psi_{\text{even/odd}}$ picture. As a result of this I think the truly qualitative advance is still not seen here and so cannot recommend to publish in Nature Communications.

Re) We appreciate the reviewer that he/she read our paper again. The reviewer said that the model described in our paper does not move significantly beyond the picture in the former work. We disagree with this criticism. In the spin rotation model reported in the former work [13], the authors considered the spin-dependent quantum interference of photoelectrons emitted from different atomic layers in the crystal, meaning that varying photon energy, synchrotron radiation for example, is required to demonstrate the spin-dependent interference. On the other hand, in our model, the spin-dependent quantum interference is based on the mirror symmetry with a fixed photon energy. We emphasize that our model would be quite general and widely applicable not only for the surface states having finite decay length into the subsurface but also two-dimensional systems where the wavefunction is purely localized in the topmost atomic layer of the crystal, metal-adsorbed semiconductor surfaces for example. We believe that our model is universal and comprehensive, and thus is significantly beyond the previous knowledge.

Reviewer #2 (Remarks to the Author):

I find that the authors have thoroughly addressed all the issues raised by all three referees. All three referees agree on the high quality of the research report in the paper. There is no correct / wrong answer when it comes to how important a paper is. For the same reason as in my first report, and also for the additional reason as in the report of the third referee, I believe the paper will draw attention of the general readership of Nature Communication; the paper will foster further research activities in the field of spin ARPES on several interesting / newly-discovered materials with strong spin-orbit coupling. For this reason, I recommend publication of this paper in current form.

Re) We are thankful to the reviewer for appreciating our work and we are happy that the reviewer feels our paper is ready for publication in Nature Communications in current form.

Reviewer #3 (Remarks to the Author):

I note that all three referees had only minor technical comments. These comments have been fully addressed by the authors in the revised version and the rebuttal letter.

The most substantial criticism was raised by referee 1, pointing out that the observation of spin-orbital texture as well as the decomposition of a spin-orbital entangled wave function into symmetric and antisymmetric parts with opposite spin are not new at all. While this is of course true, I believe that it misses the key-advance of this paper. What is new here is neither the concept of spin-orbital entanglement nor the experimental evidence for it but the development of a simple and transparent model to describe the spin polarization in photoemission from spin-orbit coupled systems. As described in my first report, I consider this a significant and timely advance.

Re) We are pleased that the reviewer appreciates our work. We also thank for pointing out the key-advance of our paper. We are delighted with the comment that the reviewer recommends publication of our paper in Nature Communications.